# Epidemiological dynamics of Ebola outbreaks

**Thomas House[1,2]***

[1]Warwick Mathematics Institute, University of Warwick, Coventry, United Kingdom; [2]Warwick Infectious Disease Epidemiology Research Centre, University of Warwick, Coventry, United Kingdom

**Abstract** Ebola is a deadly virus that causes frequent disease outbreaks in the human population. In this study, we analyse its rate of new introductions, case fatality ratio, and potential to spread from person to person. The analysis is performed for all completed outbreaks and for a scenario where these are augmented by a more severe outbreak of several thousand cases. The results show a fast rate of new outbreaks, a high case fatality ratio, and an effective reproductive ratio of just less than 1.

## Introduction

Ebola virus disease is an often fatal disease of humans that is not vaccine-preventable and has no specific treatment. A total of 25 outbreaks, believed to have arisen due to zoonotic transmission from wild mammals, have occurred since the first observed cases in humans in 1976 (*World Health Organisation, 2014a*). The current epidemic is the largest to date (*World Health Organisation, 2014b*). This gives particular urgency to quantitative estimation of epidemiological quantities relevant to Ebola, such as case fatality ratio, timing of new outbreaks, and the strength of human-to-human transmission.

The most important epidemiological quantity to estimate for an infectious disease is typically the basic reproductive ratio, $R_0$, defined as the expected number of secondary cases produced per primary case early in the epidemic (*Diekmann et al., 1990*). When $R_0$ is greater than 1, the expectation is that a new epidemic will eventually infect a significant percentage of the population if it is not stopped by interventions or chance extinction; conversely, when $R_0$ is less than 1, chance events may lead to a large number of cases, but these are always expected to be much less numerous than the total population size.

Previous attempts to estimate $R_0$ for Ebola have found values between 1.34 and 3.65 by fitting compartmental epidemic models to the incidence over time of the large outbreaks in the Democratic Republic of Congo in 1995 and Uganda in 2000 (*Chowell et al., 2004*; *Ferrari et al., 2005*; *Legrand et al., 2007*), with similar results obtained for the ongoing outbreak (*Althaus, 2014*). This leads to the question of why all completed outbreaks numbered at most several hundreds, with the typical answer being that the medical and social response to an outbreak reduces transmission, leading to an effective reproductive ratio $R_t < R_0$ (*Chowell et al., 2004*; *Legrand et al., 2007*), although it is also important to note that heterogeneity in transmission can lead to extremely high probabilities of an outbreak becoming extinct even if $R_t$ is slightly greater than 1 (*Lloyd-Smith et al., 2005*).

## Results

*Figure 1* shows the results of fitting to times between outbreaks, with *Figure 1A* showing the empirical distribution of times between outbreaks together with the fitted model distribution that has mean 1.49[1.02, 2.24] years between outbreaks and *Figure 1C* showing the posterior for the rate parameter. *Figure 1* also shows the results of fitting CFR to number of deaths and final size, with *Figure 1B* showing

*****For correspondence:**
T.A.House@warwick.ac.uk

**Competing interests:** The author declares that no competing interests exist.

**Reviewing editor**: Prabhat Jha, University of Toronto, Canada

**eLife digest** The West Africa outbreak of Ebola virus disease is larger than any of the previous outbreaks over the last four decades. Most human outbreaks likely begin when a person is infected after contact with an infected wild animal—but during an outbreak the virus can spread from person-to-person via contact with blood or other bodily fluids. There is no vaccine against Ebola nor is there a specific treatment. The percentage of infected people who have been killed by the Ebola virus in the past outbreaks varies from 50% to 90%. However, predicting how an outbreak will progress once it has started remains difficult.

For any infectious disease, it is important to estimate how many new people, on average, each person with the disease will go on to infect. When this value—called the 'basic reproductive ratio' (or $R_0$)—is greater than 1, a significant percentage of the population is expected to eventually become infected if medical interventions are not introduced. Conversely, when $R_0$ is less than 1, chance events may lead to a large number of cases, but only a fraction of the total population will be affected.

Previous estimates of the basic reproductive ratio for Ebola gave values greater than 1, making it unclear why all the completed outbreaks of Ebola had infected at most several hundred people and had not caused global pandemics. Medical intervention and control measures were generally considered the most likely answer. However, it is important to note that these previous predictions were made using data from only two large outbreaks of Ebola in 1995 and 2000.

Now, House has used a different modelling approach to estimate Ebola's reproductive ratio. The new model is based on data from for all 24 completed Ebola outbreaks and includes the time between outbreaks, the number of deaths, and the final number of cases. The model also included 'data' from a hypothetical scenario of a more severe outbreak with several thousand cases.

House revealed that new outbreaks tend to occur frequently and that often a large percentage of those infected with Ebola will die of the disease, although the exact values vary between different outbreaks. Furthermore, if there is no fundamental change compared to the past, the analysis predicts that the 'effective reproductive ratio' for person-to-person spread of Ebola (which takes into account the effect of medical intervention) is just less than 1. It also predicts that the final number of cases can be very different for different outbreaks.

House concluded that at first the current West African outbreak was unusual but still consistent with the pattern of previous outbreaks. However, as the number of people infected continued to grow, it makes this less plausible. It is now more likely that there is some fundamental difference, for example in the infectiousness of the Ebola strain, in the current outbreak compared to all the previous outbreaks; although further work would be needed to confirm this.

empirical CFRs for different outbreaks together with the fitted model distribution. Other plots in *Figure 1D,E* show the posteriors for the beta distribution parameters.

*Figure 2* shows the results of fitting to completed outbreaks, with *Figure 2A,B* giving the fitted distribution against data, *Figure 2C* showing the posterior for the reproductive ratio, which is estimated to be $R_t = 0.88[0.64, 0.96]$. *Figure 2D* shows the posterior for the geometric parameter, which is estimated to be $p = 0.089[0.029, 0.19]$.

While the model is designed not to depend explicitly on the temporal dynamics of Ebola virus disease, *Figure 3A* shows a set of 24 outbreaks simulated from a continuous-time Markov chain with the same probability distribution for final size as the estimated model. These show behaviour that is typical of near-critical branching processes, which often becoming extinct early but also often grows to significant size before extinction. *Figure 3B* plots the likelihood surface for these simulated data showing parameter identifiability.

*Figure 4* shows the results of fitting to completed outbreak final sizes augmented by an outbreak of uncertain size in the range 1000–5000. In this study, *Figure 4A* gives the fitted distribution against data, and *Figure 4B* shows the posterior for the probability of the additional outbreak, which is estimated to be $0.023[0.0015, 0.088]$. *Figure 4C* shows the posterior for the reproductive ratio, which is

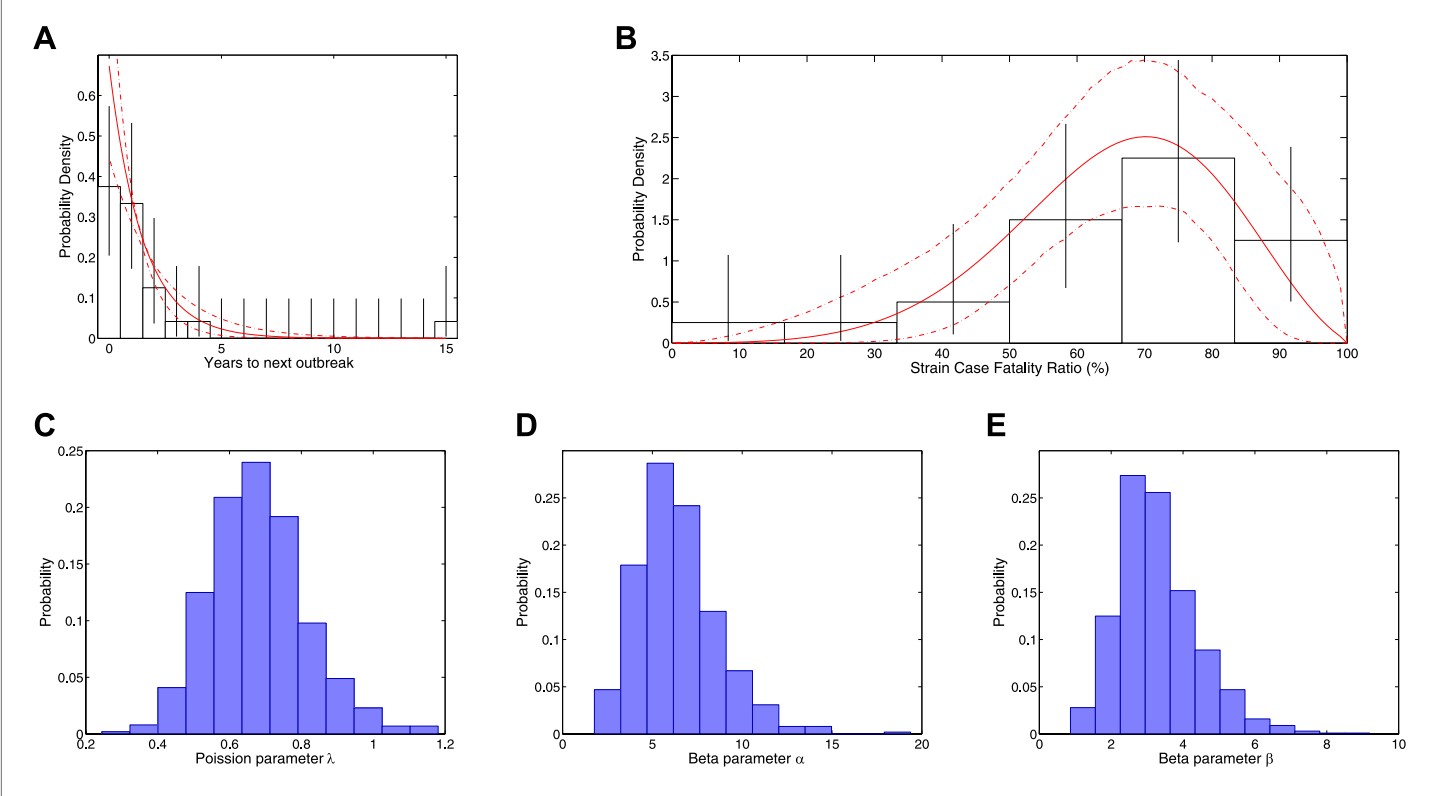

**Figure 1**. Analysis of rate of new outbreaks and case fatality ratio. **A** shows empirical data and 95% CI (black lines) together with fitted distribution and 95% CI (red lines) for rate of new outbreaks. **B** shows empirical data and 95% CI (black lines) together with fitted distribution and 95% CI (red lines) for case fatality ratio. **C** shows the posterior density for rate of new outbreaks $\lambda$, while **D** and **E** show the posterior density for the beta distribution parameters $\alpha$ and $\beta$, respectively.

estimated to be $R_t$ = 0.94[0.87, 0.99], and **Figure 4D** shows the posterior for the geometric parameter, which is estimated to be $p$ = 0.11[0.054, 0.21].

## Discussion

The results obtained point to the following conclusions about Ebola transmission dynamics. (i) The rate of new epidemics and CFR are both high, but with significant variability from outbreak to outbreak. (ii) The effective reproductive ratio $R_t$ for person-to-person transmission is just below 1. (iii) There is extremely large variability in the final size of outbreaks.

It is also important to consider the sensitivity of these conclusions. A larger final size for the current outbreak (but still significantly less than the population size of a country) as suggested by the analysis above will tend to lead to a narrower posterior about a value of $R_t$ closer to 1; this can be understood from general properties of branching processes (**Athreya and Ney, 1992**). Such a finely tuned constant value of $R_t$ would, however, become increasingly difficult to interpret as a fundamental property of the outbreak and a modelling approach in which $R_t$ was allowed to vary in time—along with the public health and behavioural responses—would be preferred.

Also, it is possible that a number of small outbreaks were not recorded by the WHO. This could be addressed through incorporation of additional variability into the model through introduction of explicit overdispersal parameters as in the study by Lloyd-Smith et al. (**Lloyd-Smith et al., 2005**) and Blumberg and Lloyd-Smith (**Blumberg and Lloyd-Smith, 2013**), although for the data currently available there was no strong evidence for overdispersal beyond that implied by the geometric distributions.

All of these conclusions suggest no reason for complacency and give support to appeals for greater resources to respond to the ongoing epidemic (**Médecins Sans Frontières, 2014**).

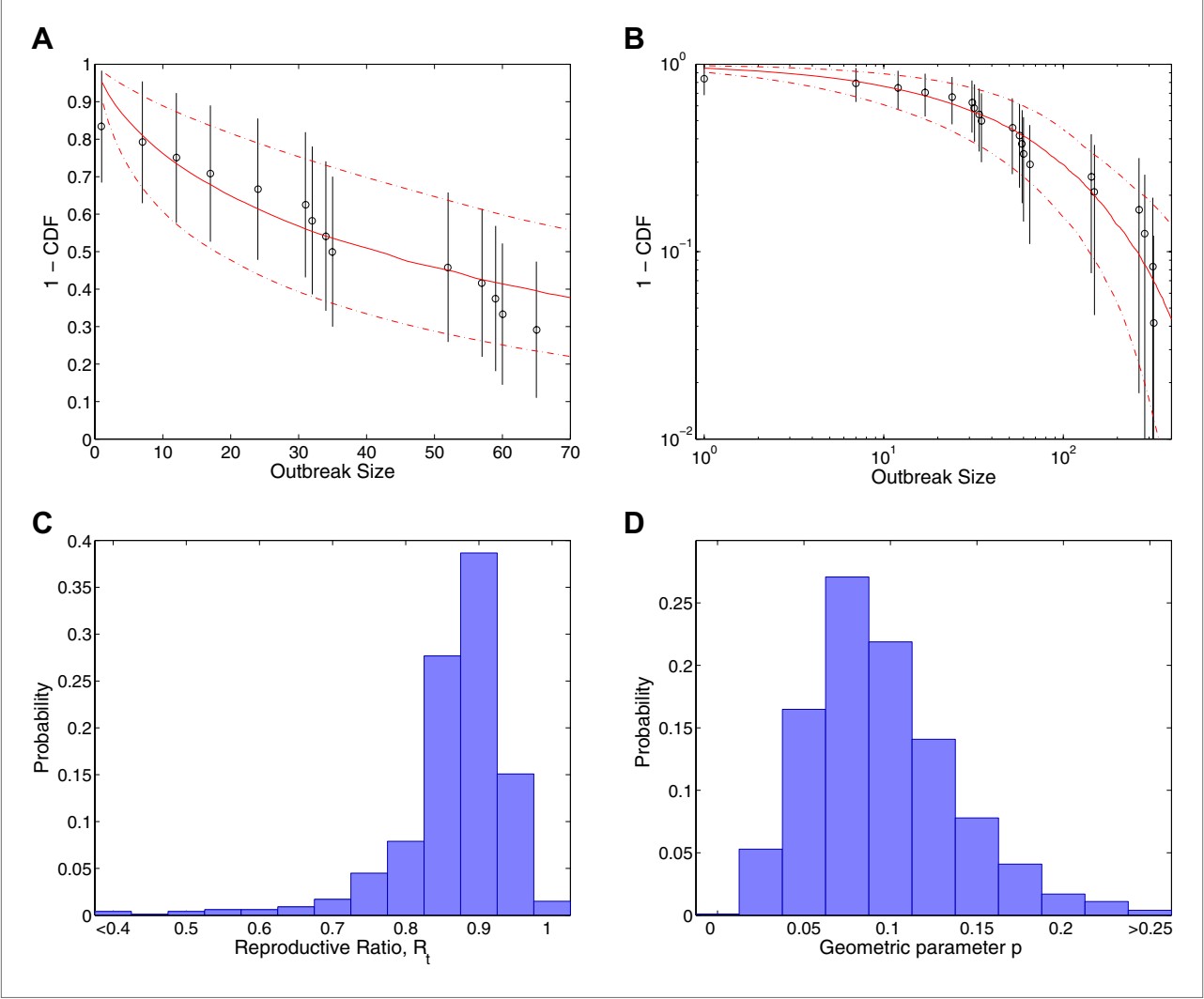

**Figure 2**. Analysis of transmission dynamics for completed outbreaks. (**A** and **B**) Model (solid red line) and 95% CI (dash-dot red line) vs data (black circles) and 95% CI (solid black lines) for different axis scales. (**C**) Posterior for values of the reproductive ratio $R_t$. (**D**) Posterior for the geometric parameter $p$.

# Materials and methods

## Description

In this study, a different approach is taken based on using the time between outbreaks, number of deaths, and final number of cases, for all 24 completed Ebola outbreaks reported by the World Health Organisation (*World Health Organisation, 2014a*). Full mathematical details of the approach are given below.

First, we model the start of new outbreaks as a 'memoryless' Poisson process with a rate λ. Secondly, we assume that each new outbreak has a case fatality ratio (CFR—the probability that a case will die) picked from a beta distribution. Thirdly, the final size model involves two components: (i) a geometrically distributed number of cases, $A$, which includes cases arising from animal-to-human and pre-control transmission; (ii) a branching process model of human-to-human transmission (*Athreya and Ney, 1992*; *Ball and Donnelly, 1995*), whose offspring distribution has mean $R_t$, generating $Z$ cases. The final size is then $K = A + Z \mid A$. This quantity should be interpreted as arising from a combination of $R_t$, $R_0$, and timing of interventions.

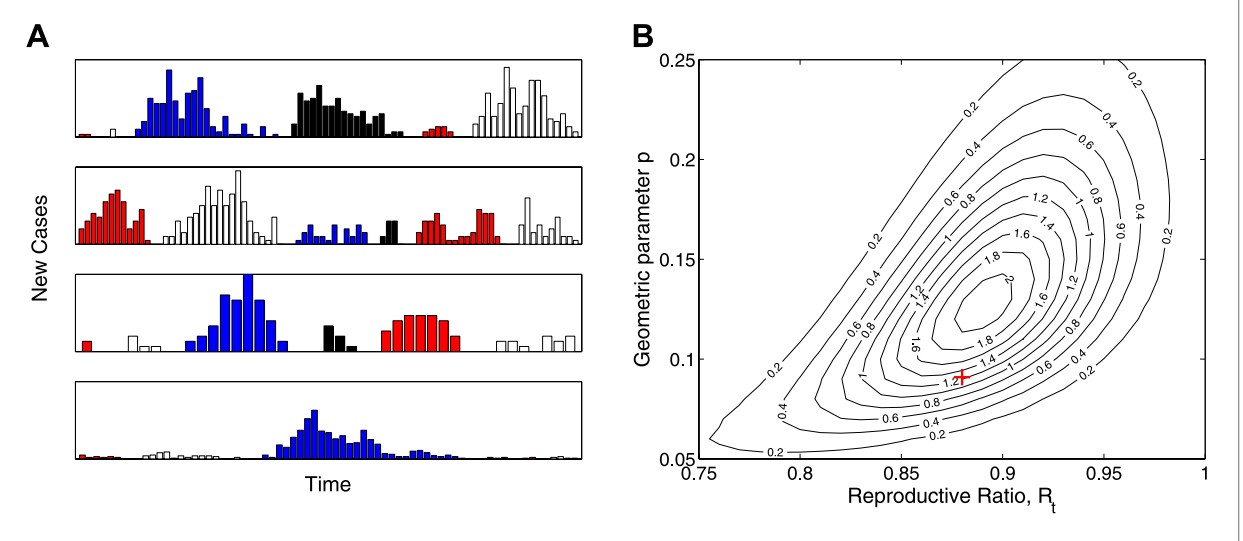

**Figure 3**. Simulation study. (**A**) Real-time model simulations, with change in colour denoting a new outbreak. (**B**) Likelihood contours (black lines and values multiplied by an unimportant constant) together with parameters used to simulate (red cross), showing that the parameters are identifiable from such data.

Bayesian MCMC with uninformative priors was used to fit all models (***Gilks et al., 1995***). Since doubts have been raised in the literature about the use of final size data for emerging diseases (***Drake, 2005***), a simulation study was also performed to test identifiability, although a recent study by Blumberg and Lloyd-Smith (***Blumberg and Lloyd-Smith, 2013***) of joint identifiability of two parameters in a related model is also highly relevant in this context.

Finally, the final size data were augmented by an outbreak of unknown size in the range 1000–5000 (with mathematical details given by ***Equation. (5)***, below) and the model was refitted. Due to the significant uncertainty in the severity of the current outbreak, this is not intended to be a real-time analysis, but rather to show how the modelling approach responds to such a scenario in general.

## Technical details

### Transmission model

Each outbreak has an initial number of cases $A$ and a secondary number of cases $Z$. The total outbreak size is $K = A + Z | A$. We model the number of initial cases as a shifted geometric distribution,

$$\Pr[A = a | p] = (1 - p)^{a-1} p. \tag{1}$$

We then model the number of secondary cases as the total progeny of a Galton–Watson branching process with $A$ initial individuals and offspring distribution given by a geometrically distributed random variable $\xi$ with mean $R_t =: (1 - q)/q$. We adapt the results from Ball and Donnelly (***Ball and Donnelly, 1995***) to our model, giving

$$\Pr[Z = z | A = a, q] = \frac{a}{a + z} \binom{2z + a - 1}{z} q^{z+a} (1 - q)^z . \tag{2}$$

This gives a formula for the total size of the outbreak of

$$\Pr[K = k | p, q] = \sum_{a=1}^{k} \Pr[A = a | p] \Pr[Z = (k - a) | A = a, q]. \tag{3}$$

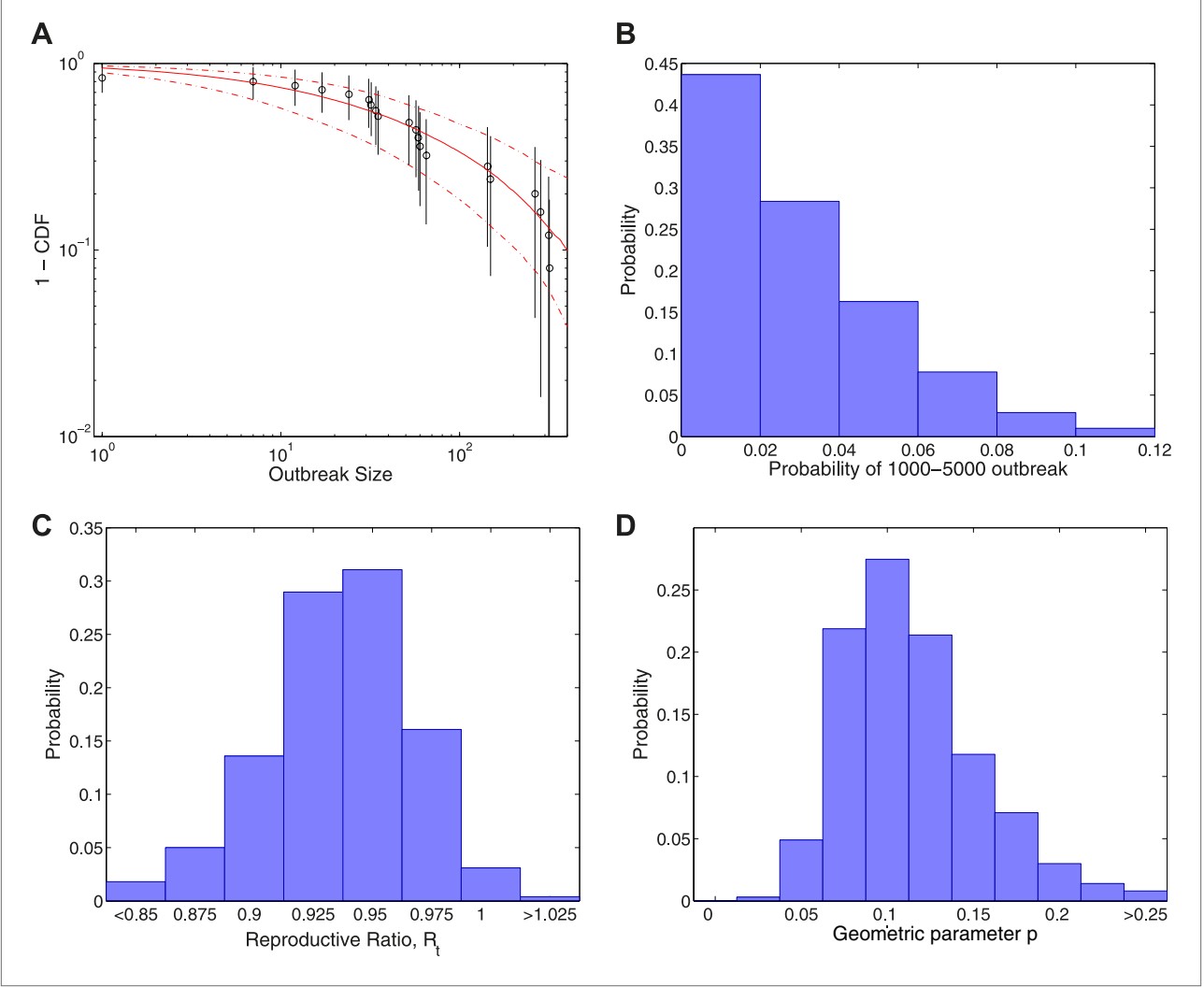

**Figure 4**. Analysis of transmission dynamics for completed outbreaks plus one outbreak of size 1000–5000. (**A**) Model (solid red line) and 95% CI (dash-dot red line) vs data (black circles) and 95% CI (solid black lines). (**B**) Posterior for the probability of the large uncertain outbreak. (**C**) Posterior for values of the reproductive ratio $R_t$. (**D**) Posterior for the geometric parameter $p$.

If the data $D$ consists of a set of $k_i$ (which is the size of outbreak $i$, with $N$ the total number of outbreaks) then the likelihood function for the transmission model is

$$L(D \mid p,q) = \prod_i \Pr[K = k_i \mid p,q].$$ (4)

When the data $D'$ consists of the set of $k_i$ augmented by an outbreak of size between $\kappa_1$ and $\kappa_2$, we use likelihood function

$$L(D' \mid p,q) = L(D \mid p,q) \sum_{k=k_1}^{k_2} \Pr[K = k \mid p,q].$$ (5)

## New outbreak model

We model the start of new outbreaks in the human population as a Poisson process of rate $\lambda$. If the time period over which $N$ outbreaks is observed is $T$ years, then the likelihood is

$$L(D \mid \lambda) = \frac{(\lambda T)^N e^{-\lambda T}}{N!}.$$ (6)

We estimate $\lambda = 0.67[0.45, 0.98]$, with posterior distribution given in **Figure 1C**. The probability density function for $t$ being the next outbreak time is

$$f(t) = \lambda e^{-\lambda t},$$

(7)

which is shown in **Figure 1A**.

## Case fatality model

We let $C_i$ be a random variable for the probability of fatality for a given case in outbreak $i$. We assume a parametric model in which this is drawn from a beta distribution, meaning that the probability density function is

$$\text{Beta}(c \mid \alpha, \beta) = \frac{c^{\alpha-1}(1-c)^{\beta-1}}{B(\alpha, \beta)}, \quad B(\alpha, \beta) := \int_0^1 x^{\alpha-1}(1-x)^{\beta-1}\,dx.$$

(8)

Then if $d_i \le k_i$ is the number of fatalities in outbreak $i$, treating each fatality as independent, conditioned on infection, gives

$$\Pr[d_i \mid k_i, \alpha, \beta] = \binom{k_i}{d_i} \frac{B(\alpha + d_i, \beta + k_i - d_i)}{B(\alpha, \beta)}.$$

(9)

Then the likelihood is

$$L(D \mid \alpha, \beta) = \prod_i \Pr[d_i \mid k_i, \alpha, \beta].$$

(10)

We estimate $\alpha = 6.1[2.8, 11]$ and $\beta = 3.1[1.5, 5.9]$, with posterior distributions given in **Figure 1D,E**.

## Statistical methodology

The MCMC methodology used was Random-walk Metropolis–Hastings with thinning to produce $10^3$ uncorrelated samples, with each posterior ultimately derived from one long chain. The parameter spaces involved are low-dimensional enough that large-scale sweeps can be performed to check for multimodality, which was not observed, and convergence of the chains was observed to be fast and independent of initial conditions.

For the simulation study, the real-time incidence curves are produced by modelling the geometric distributions as arising from Poissonian transmission with exponentially distributed rates. The times between new introductions are not explicitly modelled or shown.

## Code

MATLAB code to reproduce the analysis of this paper is available at: https://github.com/thomasallan-house/elife-ebola-code.

## Acknowledgements

Work supported by the UK Engineering and Physical Sciences Research Council. I would like to thank Deirdre Hollingsworth, Matt Keeling, and Graham Medley for helpful discussions and the Editors and Reviewers for helpful comments and suggestions.

## Additional information

### Funding

| Funder | Author |
| --- | --- |
| Engineering and Physical Sciences Research Council | Thomas House |

The funder had no role in study design, data collection and interpretation, or the decision to submit the work for publication.

### Author contributions

TH, Conception and design, Acquisition of data, Analysis and interpretation of data, Drafting or revising the article

## Additional files

### Major dataset

The following previously published dataset was used

| Author(s) | Year | Dataset title | Dataset ID and/or URL | Database, license, and accessibility information |
| --- | --- | --- | --- | --- |
| World Health Organisation | 2014 | Ebola virus disease, Fact sheet Number 103 | http://www.who.int/mediacentre/ factsheets/fs103/en/ | |

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
