## [Decision Letter]

Thank you for choosing to send your work entitled “Epidemiological Dynamics of Ebola Outbreaks” for consideration at *eLife.* Your full submission has been evaluated by Prabhat Jha (Senior editor), a Reviewing editor, and 3 peer reviewers, and the decision was reached after discussions between the reviewers. We regret to inform you that your work will not be considered further for publication in its current form.

However, given the importance of scientific debate on Ebola transmission, we would review (quickly) a substantially revised version of this paper. To consider this, we would need to be assured that you have addressed each of the substantive points below. Please also comment if you can on the specific role of any putative therapies (such as monoclonal antibodies) in affecting the R0 estimation.

The following individuals responsible for the peer review of your submission have agreed to reveal their identity: Prabhat Jha, Matt Ferrari, and Sake de Vlas.

*Reviewer 1*:

1) In general I found this to be a sound and reasonable analysis. I find that the author's assertion that R0>1 is a sufficient condition for a global pandemic to be overstated, however, and I would strongly recommend tempering the language. There are a great many conditions that contribute to the likelihood of a pandemic, and the epidemic potential, as encapsulated in R0 is only one of these a necessary condition, but hardly sufficient.

Given that the author is motivating this manuscript with the current Ebola outbreak, it would be quite interesting to comment on how likely the current outbreak (i.e. an outbreak equal to or greater than the current outbreak) would be under the author's fitted model.

*Reviewer 2*:

It is an interesting idea trying to capture the epidemiological characteristics of all observed Ebola outbreaks in just 4 parameters: p, R0 (or a function of q), α, and β. However, in my view the approach is far too simplistic. In particular, the assumption that R0 (or q) has the same value everywhere and at every time is of course incorrect. This makes the suggestion of a “global pandemic” inappropriate in the first place; the conditions that have favoured spread of Ebola in tropical Africa will never apply to most other continents.

Another problem is that the same R0 is assumed to apply throughout an outbreak. This is why the estimated posterior values of R0 are <1, for the simple reason that all observed epidemics have ended. Usually, outbreak situations are characterized by an initial value of R0 > 1 (or sometimes slightly <1), followed by a decrease through interventions, behaviour change, lack of sufficient susceptible individuals, or a combination. R0 then becomes Rt. Ebola is a terrible disease and during outbreaks people are known to change their behaviour, e.g. by avoiding contact with suspected cases. Also, isolation of patients is a very effective means of control, dramatically reducing the risk of further spread as the outbreak progresses.

Thus, these analyses need to be made more sophisticated by including heterogeneity in values of R0, between outbreaks and especially within outbreaks. Within outbreaks the value of Rt (which has value R0 at time 0) should be allowed to change (i.e. decrease), which could perhaps be done by including a time series component in the MCMC approach.

Furthermore, the author assumes the data to be perfect. However, is this true? There may have been several unnoticed outbreaks of a single to a few cases that never ended up in the WHO data base. How would this change the findings? There should be more discussion and (sensitivity) analyses about this issue.

Give the current topicality it would also be interesting to relate the ongoing very large outbreak (about 700 fatalities) to the presented analyses. Can such a large outbreak still be explained by this simple approach?

*Reviewer 3*:

Using the estimate, he concludes R0 for Ebola is less than one. I must admit, it is a simple and elegant paper, and on the face of it, it seems like a very good idea.

My chief critique is that the author has not cited and seems to be unaware of an 8-year old critique of the methodology by Drake:

(2006) The Difficulties of Predicting the Outbreak Sizes of Epidemics. PLOS Med 3(1): e23. doi:10.1371/journal.pmed.0030023

In sum, Drake concludes that in the event of a delayed onset intervention, that the final size of an epidemic could be anything. The final size is thus a poor metric for R0 estimation.

To be suitable for publication, the article would have to address the issues raised by Drake. In some sense the article would have to make the case that the model being used was appropriate to the task. The model is very simple, but Drake's point is that the final size is highly sensitive to departures from that model, and many of the modifying assumptions would likely be highly appropriate for Ebola. If a new model were adopted, the reanalysis would be extensive.

[Editors' note: further revisions were requested prior to acceptance, as described below.]

Thank you for resubmitting your work entitled “Epidemiological Dynamics of Ebola Outbreaks” for further consideration at *eLife*. Your revised article has been favorably evaluated by Prabhat Jha (Senior editor) and four reviewers. We would like to reach a final decision on this, however, and given the important public scrutiny to Ebola virus, we invite you to pay specific attention to the following in another revision of this work:

1) Provide a clearer explanation of Rt and the extent to which it fits the observed ongoing spread of Ebola (including the value of being less than <1). Several reviewers have raised specific queries on the estimation, to which we would like you to respond, specifically to those of Sake de Vlas. For example, please explain:

(a) If Ro >1, but then control is implemented and makes Re < 1, then the final size of the epidemic depends on when the control was implemented. Conversely, the final size distribution reflects Ro, Re, and timing, not just Ro. This must be discussed as a caveat to the use of final size to estimate Ro.

(b) The fixed value of Rt is rather unlikely. Still, with Rt slightly < 1, (sometimes large) outbreaks could occur, followed by 'natural' extinction. It remains an interesting theoretical exercise to demonstrate that this simplistic assumption fits to the Ebola data, including the current multi-country epidemic. Please ensure that you highlight this limitation properly in the discussion.

2) Please review each of the reviewer comments (below) and reply to me.

The next stage will involve a rapid review by the editor – Prabhat Jha – to see if the major concerns have been addressed, and if so, the paper will be accepted as submitted. If it is not, we will have to send this out for additional review. Thus, please do these replies carefully and make changes to the manuscript that you see fairly addressing these concerns.

The manuscript has been improved but there are some remaining issues that need to be addressed before acceptance, as outlined below:

*Reviewer 1*:

The author has addressed many of the concerns from the earlier reviews and I appreciate the attempts to a) temper the language, and b) include some analysis in light of the current outbreak.

I still think that the author has not described the latter analysis very well. The main text states that the data were “augmented”, though there is no real detail about what this is meant to mean (either in the main text or the appendix). I can imagine what was likely done, but the reader should have to imagine what methods were used. I would strongly suggest that the author clarify this methodology and provide some real interpretation of the output. At present, Figure 5 is presented without any real synthesis. The added paragraph in the results simply restates the figure legend without presenting any real interpretation or speculation on how to interpret these results in light of the current outbreak. I'm not suggesting that the author overreach here, but this analysis would permit some kind of statement like “if the current outbreak were to exceed ##, then there would be strong evidence of a shift in dynamics from that which resulted in the prior pattern of outbreaks.”

*Reviewer 2*: Sake de Vlas

First of all, I would like to re-emphasize that the model used for final size of Ebola outbreaks is too simplistic, or too 'general' in the author's words. Outbreaks are normally characterized by an initial value of Rt = R0 (usually >1), which reduces to a lower value (usually <1) during the epidemic, as a consequence of interventions or behaviour change of the population.

There is no reason to believe that this would be different for Ebola. On the contrary, Ebola is known to steadily spread (unnoticed) for some time, but after detection outbreaks are rapidly contained by stringent control through isolation of patients, often heroically organized by MSF. The current multi-country outbreak is characterized by a lack of trust by the affected populations, so that control cannot be organized well and transmission continues. A proper model, certainly a practical one to be used for making predictions of future course and the possible impact of interventions, should take into account such mechanisms.

Having said that, I still consider it an interesting mathematical thought experiment to study to which extent previous outbreaks (and the current one) might be explained by assuming a simple constant value of Rt. That this would result in a value of Rt slightly less than 1 is not a surprise at all, since this is the only way that outbreaks can both occur and eventually die out naturally. I remain a bit disappointed that the author did not manage to include a simple process of (e.g. logistic) decline in Rt during an epidemic, but hopefully this will be the subject of a follow-up study.

I am happy with the much improved Introduction and Discussion. The description of the Bayesian approach is adequate as well. Figure 5 is a useful addition, even though it expresses that the current massive outbreak is perceived by the model as just a rare random event. Crudely, with the estimated average 1.5 years between outbreaks and a 0.023 chance of a big one, such an outbreak is expected to occur once every 1.5/0.023 = 65 years. Would that be realistic?

All in all, I recommend acceptance of the study. Remaining comments:

1) The technical annex still mentions R0, whereas Rt would be expected.

2) Figure 4, giving an illustration of the timing of epidemics, requires some more explanation. It seems that somewhere a generation time has been assumed, since otherwise the timing of successive new cases within outbreaks cannot be related to the timing of new outbreaks.

3) Perhaps the author can calculate the occurrence of large-size outbreaks more precisely, using the detailed results of the Bayesian analyses.

*Reviewer 3*:

The response to my comment – the one dealing with the critique of methods from Drake et al. – was inadequate. In fact, I can't tell whether the author had read the whole article.

Drake's point about variability in the final size of an epidemic is a serious challenge to the validity of this work. Such critiques must be dealt with in substantive ways, and not by putting a band-aid on the analysis.

Drake's most critical point, illustrated in Figure 4, was about delayed onset of interventions. This is the most serious challenge to interpreting the analysis of final size as an estimate of R0. In fact, it seems Ebola fits this quite well, in that interventions are applied unevenly throughout the course of an epidemic, once detected, the epidemics are generally nipped in the bud and they die out. What seems much more likely to me is that R0 is higher than one before control measures are implemented, and less than one after that. In such situations, the final size of an epidemic has much more to do with the timing of the onset of interventions.

This is the point that House has not dealt with at all, as far as I can tell, and it is a serious challenge to the validity of his estimate.

*Reviewer 4:* Simon Hay

I think Thomas House has done a comprehensive job at responding to the reviewers concerns. The article is extremely brief, as per the format, but written clearly.

1) Critical arguments are around whether the concerns raised by reviewer 2 and those of Drake (2006) raised by reviewer 3, will need to be evaluated by the relevant individuals and I will be very interested in their responses.

2) Apologies if I have missed them, but there is no plot of the epidemics by size and time: that is the core dataset. This would be useful to show the reader and I imagine with a little imagination one could show all the baseline data being used. The data/code etc should also be made available on Dryad or the like.

3) When plotted elsewhere I note that there is a distinct change in the frequency of the epidemics pre and post 2000, with the frequency much greater in the last 15 years. Does this non-stationary pattern (hesitate to use the word trend) in outbreaks effect the analyses.

4) I think the impact statement should be refined. Ebola is clearly transmissible enough to cause a major epidemic – we are in the midst of one. I don't think this is exactly this is what you mean to say. Is it something more along the lines of “Despite the Rt of Ebola being less than 1, it can lead to the generation of a range of outbreak sizes consistent with the scale of the ongoing epidemic”? Regardless it needs clarifying.

---

## [Author Response]

Reviewer 1:

*1) In general I found this to be a sound and reasonable analysis. I find that the author's assertion that R0>1 is a sufficient condition for a global pandemic to be overstated, however, and I would strongly recommend tempering the language. There are a great many conditions that contribute to the likelihood of a pandemic, and the epidemic potential, as encapsulated in R0 is only one of these a necessary condition, but hardly sufficient*.

This is a fair comment and I have significantly reworded the manuscript, removing the reference to a “global pandemic” throughout.

*Given that the author is motivating this manuscript with the current Ebola outbreak, it would be quite interesting to comment on how likely the current outbreak (i.e. an outbreak equal to or greater than the current outbreak) would be under the author's fitted model*.

I have carried out significant additional MCMC runs to address this issue, leading to the results shown in Figure 5.

Reviewer 2:

*It is an interesting idea trying to capture the epidemiological characteristics of all observed Ebola outbreaks in just 4 parameters: p, R0 (or a function of q), α, and β. However, in my view the approach is far too simplistic. In particular, the assumption that R0 (or q) has the same value everywhere and at every time is of course incorrect. This makes the suggestion of a “global pandemic” inappropriate in the first place; the conditions that have favoured spread of Ebola in tropical Africa will never apply to most other continents*.

*Another problem is that the same R0 is assumed to apply throughout an outbreak. This is why the estimated posterior values of R0 are <1, for the simple reason that all observed epidemics have ended. Usually, outbreak situations are characterized by an initial value of R0 > 1 (or sometimes slightly <1), followed by a decrease through interventions, behaviour change, lack of sufficient susceptible individuals, or a combination. R0 then becomes Rt. Ebola is a terrible disease and during outbreaks people are known to change their behaviour, e.g. by avoiding contact with suspected cases. Also, isolation of patients is a very effective means of control, dramatically reducing the risk of further spread as the outbreak progresses*.

*Thus, these analyses need to be made more sophisticated by including heterogeneity in values of R0, between outbreaks and especially within outbreaks. Within outbreaks the value of Rt (which has value R0 at time 0) should be allowed to change (i.e. decrease), which could perhaps be done by including a time series component in the MCMC approach*.

I am in general agreement with the spirit of these points. I now discuss R_T_ rather than R_0_, and have re-moved discussion of a “global pandemic”. I am in slight disagreement with the reviewer on two technical matters: (1) If a pattern of minor outbreaks is really consistent with a supercritical branching process model (which it may well be) then there will in fact be some posterior mass in that region of parameter space. (2) A discrete-time stochastic model like a Galton Watson process can be a good approximation to the final outcome of a great many real-time epidemic models as seen through coupling arguments – the standard stochastic SIR and SEIR models are just two that will lead to exactly the model considered, but a great many others will be well approximated by it.

The model proposed is not intended as a fully mechanistic approach; instead I hope to capture a small number of key mechanisms in a principled way, while still being able to implement a full likelihood-based fitting procedure without any methods such as fixing parameters by hand.

*Furthermore, the author assumes the data to be perfect. However, is this true? There may have been several unnoticed outbreaks of a single to a few cases that never ended up in the WHO data base. How would this change the findings? There should be more discussion and (sensitivity) analyses about this issue*.

This is now considered in the Discussion section; since the models involved are reasonably transparent I argue that it is clear how they would respond to such a scenario.

*Give the current topicality it would also be interesting to relate the ongoing very large outbreak (about 700 fatalities) to the presented analyses. Can such a large outbreak still be explained by this simple approach*?

I have carried out significant additional MCMC runs to address this issue, leading to the results shown in Figure 5.

Reviewer 3:

*Using the estimate, he concludes R0 for Ebola is less than one. I must admit, it is a simple and elegant paper, and on the face of it, it seems like a very good idea*.

*My chief critique is that the author has not cited and seems to be unaware of an 8-year old critique of the methodology by Drake*:

*(2006) The Difficulties of Predicting the Outbreak Sizes of Epidemics*. *PLOS Med 3(1): e23.*
*doi:10.1371/journal.pmed.0030023*

*In sum, Drake concludes that in the event of a delayed onset intervention, that the final size of an epidemic could be anything. The final size is thus a poor metric for R0 estimation*.

*To be suitable for publication, the article would have to address the issues raised by Drake. In some sense the article would have to make the case that the model being used was appropriate to the task. The model is very simple, but Drake's point is that the final size is highly sensitive to departures from that model, and many of the modifying assumptions would likely be highly appropriate for Ebola. If a new model were adopted, the reanalysis would be extensive*.

I have now performed a simulation study to address this issue. As I interpret it, Drake’s argument is that there can be significant variability in the distribution of the final size K|R_0_, which has implications for predictability. Given N observed samples from such a distribution, however, the posterior distribution for R_0_|{K_I_}^N^_I=1_ need not be excessively wide, as the simulation study indeed shows. I also make reference to the recent study by Blumberg and Lloyd-Smith that deals with these issues for a related model in more detail. As I argue above, the simplicity of the model (which I would prefer to call ‘generality’ but of course I’d say that) is a strength in this context since it can be fitted with full statistical rigour and approximates the behaviour of many other models.

[Editors' note: further revisions were requested prior to acceptance, as described below.]

*1) Provide a clearer explanation of Rt and the extent to which it fits the observed ongoing spread of Ebola (including the value of being less than <1)*. *Several reviewers have raised specific queries on the estimation, to which we would like you to respond, specifically to those of Sake de Vlas. For example, please explain:*

(a) If Ro >1, but then control is implemented and makes Re < 1, then the final size of the epidemic depends on when the control was implemented. Conversely, the final size distribution reflects Ro, Re, and timing, not just Ro. This must be discussed as a caveat to the use of final size to estimate Ro.

I have added the following sentence to the second paragraph of the Methods, about the final outbreak size random variable: “This quantity should be interpreted as arising from a combination of R_T_, R_0_ and timing of interventions.”

*(b) The fixed value of Rt is rather unlikely. Still, with Rt slightly < 1, (sometimes large) outbreaks could occur, followed by 'natural' extinction. It remains an interesting theoretical exercise to demonstrate that this simplistic assumption fits to the Ebola data, including the current multi-country epidemic. Please ensure that you highlight this limitation properly in the discussion*.

The following sentence has been added to the second paragraph of the Discussion: “Such a finely tuned constant value of R_T_ would, however, become increasingly difficult to interpret as a fundamental property of the outbreak and a modelling approach in which R_T_ was allowed to vary in time – along with the public health and behavioural responses – would be preferred.”

Reviewer 1:

*The author has addressed many of the concerns from the earlier reviews and I appreciate the attempts to a) temper the language, and b) include some analysis in light of the current outbreak*.

*I still think that the author has not described the latter analysis very well. The main text states that the data were “augmented”, though there is no real detail about what this is meant to mean (either in the main text or the appendix). I can imagine what was likely done, but the reader should have to imagine what methods were used. I would strongly suggest that the author clarify this methodology and provide some real interpretation of the output. At present, Figure 5 is presented without any real synthesis. The added paragraph in the results simply restates the figure legend without presenting any real interpretation or speculation on how to interpret these results in light of the current outbreak. I'm not suggesting that the author overreach here, but this analysis would permit some kind of statement like “if the current outbreak were to exceed ##, then there would be strong evidence of a shift in dynamics from that which resulted in the prior pattern of outbreaks*.*”*

The method of augmentation is now given in the Methods, in particular [Disp-formula equ5].

The additional sentence in the Discussion section detailed above should address this issue; I am not sure how to quantify the appropriate number, but at some point the model becomes unnatural and the solution to this is likely to be time-inhomogeneity as suggested by the reviewers.

Reviewer 2*: Sake de Vla*

*First of all, I would like to re-emphasize that the model used for final size of Ebola outbreaks is too simplistic, or too 'general' in the author's words. Outbreaks are normally characterized by an initial value of Rt = R0 (usually >1), which reduces to a lower value (usually <1) during the epidemic, as a consequence of interventions or behaviour change of the population*.

*There is no reason to believe that this would be different for Ebola. On the contrary, Ebola is known to steadily spread (unnoticed) for some time, but after detection outbreaks are rapidly contained by stringent control through isolation of patients, often heroically organized by MSF. The current multi-country outbreak is characterized by a lack of trust by the affected populations, so that control cannot be organized well and transmission continues. A proper model, certainly a practical one to be used for making predictions of future course and the possible impact of interventions, should take into account such mechanisms*.

*Having said that, I still consider it an interesting mathematical thought experiment to study to which extent previous outbreaks (and the current one) might be explained by assuming a simple constant value of Rt. That this would result in a value of Rt slightly less than 1 is not a surprise at all, since this is the only way that outbreaks can both occur and eventually die out naturally. I remain a bit disappointed that the author did not manage to include a simple process of (e.g. logistic) decline in Rt during an epidemic, but hopefully this will be the subject of a follow-up study*.

*I am happy with the much improved Introduction and Discussion. The description of the Bayesian approach is adequate as well. Figure 5 is a useful addition, even though it expresses that the current massive outbreak is perceived by the model as just a rare random event. Crudely, with the estimated average 1.5 years between outbreaks and a 0.023 chance of a big one, such an outbreak is expected to occur once every 1.5/0.023 = 65 years. Would that be realistic*?

I understand the reviewer’s disappointment, but the issue here is really one of lack of data availability and quality for a temporal analysis, together with issues of mathematical formulation and fitting of a more complex model, that will require a substantial additional study to address adequately.

Nevertheless, the reviewer’s general characterisation of the study is I believe fair; something more complex would be needed to inform predictions and intervention strategies, but I would argue that such a model should incorporate insights from analysis of previous outbreaks and it is this prior information that I seek to provide here.

1) The technical annex still mentions R0, whereas Rt would be expected.

This has been modified to read R_T_.

2) Figure 4, giving an illustration of the timing of epidemics, requires some more explanation. It seems that somewhere a generation time has been assumed, since otherwise the timing of successive new cases within outbreaks cannot be related to the timing of new outbreaks.

This is now detailed in the Methods section.

3) Perhaps the author can calculate the occurrence of large-size outbreaks more precisely, using the detailed results of the Bayesian analyses.

This is possible; however the estimates quickly become dominated by uncertainty in parameters.

Reviewer 3:

*The response to my comment* – *the one dealing with the critique of methods from Drake et al.* – *was inadequate. In fact, I can't tell whether the author had read the whole article*.

*Drake's point about variability in the final size of an epidemic is a serious challenge to the validity of this work. Such critiques must be dealt with in substantive ways, and not by putting a band-aid on the analysis*.

*Drake's most critical point, illustrated in*
Figure 4*, was about delayed onset of interventions. This is the most serious challenge to interpreting the analysis of final size as an estimate of R0. In fact, it seems Ebola fits this quite well, in that interventions are applied unevenly throughout the course of an epidemic, once detected, the epidemics are generally nipped in the bud and they die out. What seems much more likely to me is that R0 is higher than one before control measures are implemented, and less than one after that. In such situations, the final size of an epidemic has much more to do with the timing of the onset of interventions*.

*This is the point that House has not dealt with at all, as far as I can tell, and it is a serious challenge to the validity of his estimate*.

I believe that I have understood Drake’s point; however the reviewer has not replied to my initial response to this objection, which I believe remains valid. This was essentially that (i) regardless of temporal behaviour, the final size of an outbreak should still follow *some* distribution, and (ii) sufficient samples from this distribution turn out to enable parameter identifiability for the model currently under consideration. Establishing (ii) required significant additional work as presented in the revised paper, which I believe does represent a substantive attempt to address the concerns.

Reviewer 4*: Simon Hay*

*I think Thomas House has done a comprehensive job at responding to the reviewers concerns. The article is extremely brief, as per the format, but written clearly*.

*1) Critical arguments are around whether the concerns raised by reviewer 2 and those of Drake (2006) raised by reviewer 3, will need to be evaluated by the relevant individuals and I will be very interested in their responses*.

*2) Apologies if I have missed them, but there is no plot of the epidemics by size and time: that is the core dataset. This would be useful to show the reader and I imagine with a little imagination one could show all the baseline data being used. The data/code etc should also be made available on Dryad or the like*.

The data on which the analysis is based are shown in black in Figure 1 2A,B and 4A – since the raw dataset is owned by WHO I am unsure about release; however I will of course provide access to the code (https://github.com/thomasallanhouse/elife-ebola-code) and once the code is of sufficient quality, I will release it more generally via the EpiStruct project.

*3) When plotted elsewhere I note that there is a distinct change in the frequency of the epidemics pre and post 2000, with the frequency much greater in the last 15 years. Does this non-stationary pattern (hesitate to use the word trend) in outbreaks effect the analyses*.

Figure 1 shows that on aggregate, the pattern of epidemics is consistent with an Exponential/memory less distribution; the question of testing for clustering is an interesting one, however it is arguably beyond the scope of the current work.

*4) I think the impact statement should be refined. Ebola is clearly transmissible enough to cause a major epidemic* – *we are in the midst of one. I don't think this is exactly this is what you mean to say. Is it something more along the lines of “Despite the Rt of Ebola being less than 1, it can lead to the generation of a range of outbreak sizes consistent with the scale of the ongoing epidemic”? Regardless it needs clarifying*.

This was an unfortunate error, I did not notice the Impact Statement during resubmission, and will modify to “The pattern of past Ebola outbreaks is indicative of an effective reproductive ratio of less than 1, which can lead to the generation of a range of outbreak sizes consistent with the scale of the ongoing epidemic”.